# Low-Pb High-Piezoelectric Ceramic System (1−x)Ba(Zr_0.18_Ti_0.82_)O_3_–x(Ba_0.78_Pb_0.22_)TiO_3_

**DOI:** 10.3390/ma15144760

**Published:** 2022-07-07

**Authors:** Chao Zhou, Jiajing Li, Xiaoxiao Zhang, Tiantian Yu, Yin Zhang, Sen Yang

**Affiliations:** MOE Key Laboratory for Nonequilibrium Synthesis and Modulation of Condensed Matter, School of Physics, Xi’an Jiaotong University, Xi’an 710049, China; liag120@stu.xjtu.edu.cn (J.L.); zhangxiaoxiao@stu.xjtu.edu.cn (X.Z.); ytt19980807@stu.xjtu.edu.cn (T.Y.); yzhang18@xjtu.edu.cn (Y.Z.)

**Keywords:** low-Pb, piezoelectric, morphotropic phase boundary, phase transition

## Abstract

Piezoelectric materials, especially Pb-based piezoelectric materials, are widely used in the key components of sensors, actuators, and transducers. Due to the rising concern of the toxicity of Pb, global legislation has been adopted to restrict the use of Pb. Given that the available Pb-free piezoelectric materials cannot replace the Pb-based ones for various reasons, we designed and fabricated a low-Pb piezoelectric solid-solution ceramic system, (1–x)Ba(Zr_0.18_Ti_0.82_)O_3_–x(Ba_0.78_Pb_0.22_)TiO_3_ (denoted as BZ_0.18_T–xBP_0.22_T herein). The crystal structure, ferroelectric, dielectric, and piezoelectric properties of the BZ_0.18_T–xBP_0.22_T system were systematically studied. With the increase in BP_0.22_T content, the structure of the samples changed from a rhombohedral phase to a tetragonal phase; the intermediate composition x = 0.5 was located at the morphotropic phase boundary of the system and corresponded to the state with the coexistence of the rhombohedral and tetragonal phases. Moreover, x = 0.5 exhibited the optimum comprehensive properties among all the samples, with a piezoelectric coefficient d_33_ of 240 pC/N, a maximum dielectric temperature Tm of 121.1°C, and a maximum polarization Pm of 15 μC/cm^2^. Our work verifies the validity of the route to design low-Pb high-piezoelectric materials and may stimulate the interests for exploring new low-Pb high-performance ferroelectric and piezoelectric materials.

## 1. Introduction

Piezoelectric materials can realize the conversion between elastic energy and electric energy; thus, they are widely used in actuators, sensors, and transducers [1]. Since the discovery of high piezoelectricity in Pb(Zr_0.52_Ti_0.48_)O_3_ (PZT) [2], Pb-based piezoelectric materials, i.e., Pb(Mg_1/3_Nb_2/3_)O_3_-PbTiO_3_ (PMN-PT) [3] and Pb(Zn_1/3_Nb_2/3_)O_3_-PbTiO_3_ (PZN-PT) [4], have accounted for the majority of the market share [5].

However, with the rising concern regarding the hazards for both human health and the environment caused by the toxic element Pb, the use of Pb in electronic devices is facing fierce circumstance and will be rigidly limited by laws and regulations, such as “Waste Electrical and Electronic Equipment” (WEEE) and “The Restriction of the use of certain Hazardous substances in Electrical and Electronic Equipment” (RoHS). In such a situation, high-performance Pb-free or low-Pb piezoelectric materials are highly desired. In the last two decades, many Pb-free piezoelectric materials have been introduced, such as (K_0.5_Na_0.5_)NbO_3_ (KNN) [6], (Bi_1/2_Na_1/2_)TiO_3_-BaTiO_3_ (BNT-BT) [7], Ba(Zr_0.2_Ti_0.8_)O_3_–x(Ba_0.7_Ca_0.3_)TiO_3_ (BZTBCT) [8], etc. Although much effort has been devoted to this task, it has to be admitted that all of the above-mentioned Pb-free materials face various problems; for example, the raw materials (K_2_CO_3_, Na_2_CO_3_) used to fabricate KNN are moisture-sensitive [9], the piezoelectricity of BNT-BT is inferior to most Pb-based materials [10], and the maximum dielectric temperature (Tm) is below 100°C (Tm of the optimum composition BZT-0.5BCT is 93°C) [8]. As a result, none of the above Pb-free materials can substitute for Pb-based ones. 

Therefore, in the present study, we explore low-Pb high-piezoelectric materials as an alternative. As a comprehensive consideration, such a strategy will not only help people confront the situation of the strict limits on the use of Pb, it will also gain time for the exploration of Pb-free high-performance piezoelectric materials that can completely replace the Pb-based ones. 

It has been well acknowledged that the design of morphotropic phase boundary (MPB) is an effective route to obtain high piezoelectricity [11]. At MPB, the coexistence of rhombohedral (*R*) and tetragonal (*T*) phases leads to the facilitated rotation and switching of polarization under external stress, thus yielding a high piezoelectric response [8,12]. The end members of such systems should meet the following criteria: (1) one end member undergoes the transition from the cubic (*C*) to *T* phase, while the other end member undergoes the transition from the *C* to *R* phase; (2) the two end members can form a solid-solution compound; then, a triple-point would exist [13]. We discovered that Zr-doped BaTiO_3_ [14] and Pb-doped BaTiO_3_ [15], satisfy the above requirements. 

For the end member Zr-doped BaTiO_3_, i.e., Ba(Zr_m_Ti_1__–__m_)O_3_ (BZT), to ensure it satisfies the abovementioned criteria, m was fixed as 0.18. For the end member Pb-doped BaTiO_3_, i.e., (Ba_1__–__n_Pb_n_)TiO_3_ (BPT), the value of n should guarantee that the BPT undergoes one single transition from a cubic to a tetragonal phase. According to the phase diagram [15], when n exceeds 0.2, this occurs. Therefore, the value of n that satisfies the criterion is not unique and was selected as 0.22 in the present study. However, the value of n should not be too large, which would be contrary to our aim—“low Pb” content. Thus, based on the two end member compounds, we designed the low-Pb solid-solution system (1–x)Ba(Zr_0.18_Ti_0.82_)O_3_–x(Ba_0.78_Pb_0.22_)TiO_3_. 

## 2. Materials and Methods

The (1–x)Ba(Zr_0.18_Ti_0.82_)O_3_–x(Ba_0.78_Pb_0.22_)TiO_3_ (denoted as BZ_0.18_T–xBP_0.22_T hereafter; x represents the mole fraction of (Ba_0.78_Pb_0.22_)TiO_3_) ceramic samples were prepared with the conventional solid-state reaction method, with the raw chemicals of BaCO_3_ (99.95%), ZrO_2_ (98%), TiO_2_ (99.9%), and PbO (99.9%). Considering that a Pb-induced nonstoichiometric compound usually leads to the formation of an oxygen vacancy [16], excess PbO (5 at%) was added to compensate for the loss of Pb. The calcining was performed at 1200°C in air for 2 h, and the sintering was performed at 1250°C in air for 3 h. The crystal structure was detected using an X-ray diffractometer (XRD, Bruker D8 ADVANCE, Hamburg, Germany). The temperature spectrum of dielectric permittivity was measured using an LCR meter (HIOKI-3532) at 10 kHz in a temperature chamber (Linkam), with the ramp rate of 2°C/min. The ferroelectric properties (Polarization-Electric field loops) were measured using a ferroelectric tester (Radiant Workstation Premier II, Albuquerque, NM, USA) at a frequency of 10 Hz at room temperature (22°C). The piezoelectric coefficient d_33_ was measured using a Berlincourt-type d_33_ meter (ZJ-3A). Before piezoelectric measurements, the ceramic samples were poled at 22°C for 1.5 h under an electric field of 1.0 kV/mm. 

## 3. Results and Discussion

### 3.1. Structural Characterization

Figure 1a shows the X-ray diffraction (XRD) patterns for all the BZ_0.18_T–xBP_0.22_T samples. No reflections of a second phase appeared, demonstrating that all the samples possessed a pure perovskite ABO_3_ structure [17]. The XRD profile of (200) reflection is shown in Figure 1b. From x = 0 to 0.3, with the increase in BP_0.22_T content, the reflection of (200) moved toward a higher angle, revealing the lattice constant decreased, which was caused by the smaller ionic radius of Pb^2+^ [15]. No splitting of the (200) reflection suggested a rhombohedral crystal structure [17]. On the other hand, the (200) reflection moving toward a higher angle also implied that the BZ_0.18_T and BP_0.22_T formed into solid-solution; otherwise, the BP_0.22_T would distribute mainly at the grain boundary and would not change the lattice parameters [18]. From x = 0.6 to 1.0, the reflection of the (200) split into two peaks, agreeing well with the detected tetragonal crystal structure in Pb-doped BaTiO_3_ [15]. By contrast, the intermediate composition, x = 0.5, corresponded to the state with the coexistence of *R* and *T* phases, coined the MPB composition [8]. 

### 3.2. Dielectric Permittivity versus Temperature Spectrum

Based on the XRD results in Figure 1, several typical compositions were selected for further investigation, x = 0, 0.1, 0.5, and 0.7. Figure 2 shows the temperature spectrum of the dielectric permittivity (ε at 10kHz) for the selected typical compositions (for ε measured at multiple frequencies, please refer to Appendix A). For x = 0.0 (BZ_0.18_T), which exhibited an *R* phase at room temperature, there appeared only one dielectric peak at 56.6°C, indicating the transition from the *C* to *R* phase and agreed well with a previous report [14]; for x = 0.7, which exhibited a *T* phase at room temperature, there also appeared only one peak indicating the transition from the *C* to *T* phase [15]. For the intermediate compositions x = 0.1 and x = 0.5, there appeared two dielectric peaks, of which the one at a higher temperature indicated the *C*-*T* transition, and the other one at a lower temperature indicated the *T*-*R* transition. The doped BP_0.22_T increased both the maximum dielectric temperature (Tm) and the *T*-*R* transition temperature. Compared with the other compositions, x = 0.5 possessed both relatively large dielectric permittivity and good temperature stability. The observed composition-dependent dielectric behavior was consistent with those of the previously reported ferroelectric MPB systems, i.e., BZT-BCT [8] and BHT-BCT [13]. 

Compared with a normal paraelectric–ferroelectric transition [15], the shapes of the dielectric peak at Tm for the BZ_0.18_T–xBP_0.22_T samples expanded to a broader temperature range, and this is the typical feature of a diffuse phase transition [19]. In the paraelectric phase, the inverse dielectric permittivity (1/ε) for ferroelectric materials obeys the Curie–Weiss Law [19]: (1)$$\frac{1}{\varepsilon }=\frac{T-T_{C}}{C}$$
where T_C_ is the Curie temperature, and C is the Curie–Weiss constant. 

The inverse dielectric permittivity versus temperature curves of the typical compositions x = 0, 0.1, 0.5, and 0.7 are depicted in Figure 3. To investigate the diffuse phase transition behaviors of the BZ_0.18_T–xBP_0.22_T samples, the modified Curie–Weiss Law was employed [20]:(2)$$\frac{1}{\varepsilon }-\frac{1}{\varepsilon_{m}}=\frac{{\left(T-T_{m}\right)}^{\gamma }}{C}$$
where γ denotes the degree of diffusion constant; γ = 1 indicates the normal ferroelectric transition behavior, and γ = 2 indicates the complete diffuse phase transition behavior, commonly regarded as relaxor behavior [19]. 

The plots of ln($\frac{1}{\varepsilon }-\frac{1}{\varepsilon_{m}}$) versus ln($T-T_{m}$) curves are shown in the insets of Figure 3. The fitted parameters, T_C_, T_m_, and C and the estimated values of γ for x = 0, 0.1, 0.5, and 0.7 are listed in Table 1. 

**Figure d64e681:**
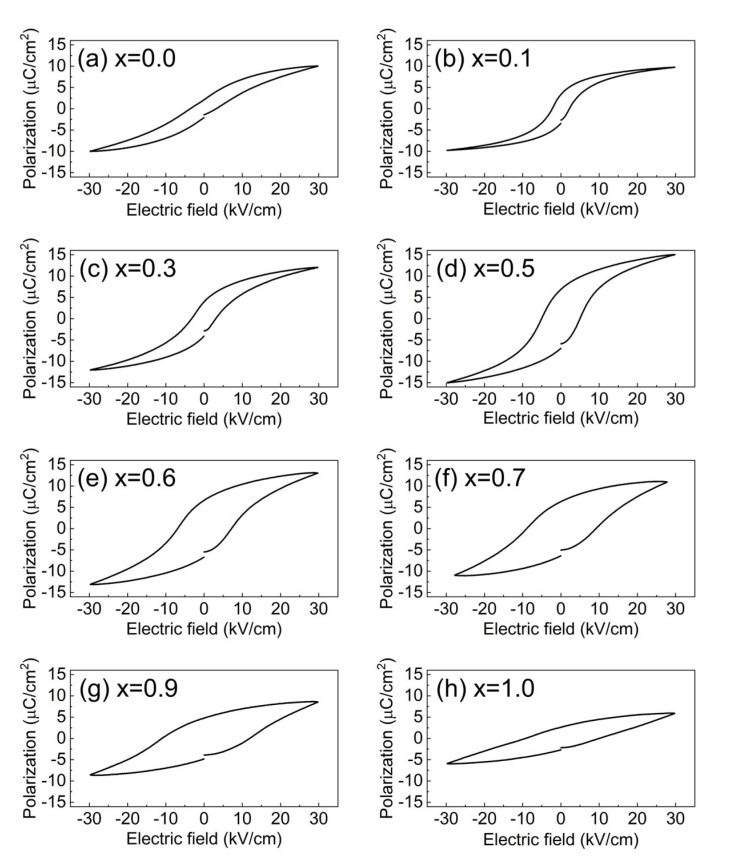


The composition x = 0.1 exhibited the strongest diffuse phase transition behavior, which can be understood from two aspects: (1) on one hand, as can be seen in Figure 2, the transition peak corresponded to the convergence of the *C*-*T* and *T*-*R* transitions; (2) on the other hand, for the solid-solution system (1−x)BZT–xBPT, while the content of the BPT was low (i.e., 0 < x ≤ 0.1), the solution can be regarded as Pb and Ti co-doped BZT. Although Pb^2+^ and Ba^2+^ are of equal valence, their ionic radii are different. Substitution of Zr^4+^ with Ti^4+^ is of the same type. Therefore, co-doping of Pb^2+^ and Ti^4+^ would probably result in a relaxor-like state [21]. 

It should be noted that the Tm of the MPB composition of the BZ_0.18_T–xBP_0.22_T system, x = 0.5, was 121.1°C, nearly thirty degrees higher than that of the famous MPB composition BZT-0.5BCT (93°C) [8]. This means that the upper limit of working temperature for BZ_0.18_T-0.5BP_0.22_T is higher than that of BZT-0.5BCT, proving the validity of designing low-Pb high-piezoelectric materials via the MPB mechanism.

### 3.3. Ferroelectric and Piezoelectric Properties

The polarization (P)—ferroelectric field (E) hysteresis loops are shown in Figure 4. With the increase in BP_0.22_T content from x = 0 to x = 0.5, the maximum polarization Pm and remnant polarization Pr first increased and then decreased, both reaching the maximum value at x = 0.5, which is clearly seen in Figure 5a,b, respectively. Generally, the enhancement of polarization results mainly from two factors: (1) the increase in the intrinsic spontaneous polarization [22] and (2) the facilitation of the ferroelectric domain switching [23]. As for the BZ_0.18_T–xBP_0.22_T system, both factors may contribute to the observed increase in polarization. On one hand, the increasing content of BP_0.22_T is essentially equal to the increasing substitution of BaTiO_3_ by PbTiO_3_. PbTiO_3_ possesses a larger c/a ratio, which results in a larger ionic polarization [24]; so the introduction of BP_0.22_T (within a certain amount) leads to the increase in intrinsic spontaneous polarization [15]. On the other hand, when the volume fraction of BP_0.22_T reaches a certain content, as 50 at% (x = 0.5) in the present study, the material locates at the MPB that corresponds to the state where the *R* phase and *T* phase coexist. The thermodynamic state at MPB has been well illustrated as the flattening of the Landau energy landscape with near-zero energy barrier between different neighboring phases [25,26]. At MPB, the ferroelectric domain can be switched under a lower field (although the coercive field E_C_ of x = 0.5 was not the minimum among all the compositions) [12]. Therefore, the maximum polarization was obtained at x = 0.5. From x = 0.5 to 1.0, the greater potential well depths of PbTiO_3_ gradually dominated and caused the domain switching to become more difficult as evidenced by the changing trend of E_C_ (Figure 5c) [27]; so, the polarization decreased monotonously as the content increased from x = 0.5 to 1.0.

As for the minimum E_C_ appearing at x = 0.3, it was probably due to the observed “invisible phase boundary” in such MPB-involved systems [28]. For the triple-point MPB involved system, the MPB derives from its starting point—the triple-point where three phases coexist [13]. Such phase coexistence naturally gives rise to the minimum energy barrier within the whole diagram; so, the MPB that derives from it yields a large response under small external fields. The minimum E_C_ appearing at x = 0.3 suggested that the triple point of the system corresponded to the composition x = 0.3, similar to a previously reported (1−x)Ba(Ti_0.8_Hf_0.2_)O_3_–x(Ba_0.7_Ca_0.3_)TiO_3_ system [28,29]. In addition, from x = 0.3 to 1.0, E_C_ increased monotonously; this phenomenon was ascribed to the abovementioned greater potential well depths of PbTiO_3_ [30].This could also explain the difference between the BZ_0.18_T–xBP_0.22_T system and the BZT-BCT system; for the BZT-BCT, the lowest E_C_ was observed at the MPB composition BZT-0.5BCT [8]. 

The measured piezoelectric coefficients d_33_ of all the samples are shown in Figure 5d. Among all the available compositions, there were two peak values, one at x = 0.1 and the other at x = 0.5. The physical basis of the enhanced piezoelectricity at the MPB is the unique thermodynamic state with low energy barriers between neighboring ferroelectric states [12]. x = 0.5 was the MPB composition of the system, corresponding to the state of two-phase coexistence with a low energy barrier, which endowed it with a relatively low E_C_ and high polarization, as well as transverse instability under external stimuli [26], thus exhibiting a large d_33_ value (240 pC/N), even exceeding that of the undoped PZT ceramics (160~190 pC/N) [31]. It should be emphasized that two-phase coexistence does not guarantee high piezoelectricity, although enhancement of piezoelectricity is usually observed, as evidenced in BNT-BT materials [10]. The increase in d_33_ of x = 0.5 compared with other off-MPB compositions originates from the thermodynamic feature of MPB, which inherits partly from the triple point where three phases coexist with near-zero energy barriers. Therefore, in principle, the energy barrier of the MPB is lower than that of the polymorphic boundary (PB), and naturally the piezoelectric properties at the MPB are superior to those at the PB [32]. 

It is interesting that when x = 0.1, neither the end member composition nor the composition at phase boundary also showed a peak value of d_33_ of 186 pC/N, comparable to the performance of undoped PZT ceramic. This should be attributed to the strongest diffuse phase transition behavior detected for the composition x = 0.1 (Table 1), as discussed above. In fact, tuning the normal ferroelectric transition to a diffuse phase transition has been regarded one of the efficient routes to obtain large piezoelectricity [33,34]. Overall, the composition x = 0.5 exhibited the optimum comprehensive properties regarding the polarization, the coercive field, and the piezoelectricity. 

Finally, it should be noted that Pb has an increased migration ability during the formation of ferroelectric materials with their content, thus affecting the properties and microstructure of the samples [35,36,37]. Although the Pb-induced nonstoichiometric and imperfection factors are beyond the scope of the current work, this issue should and will be seriously considered in the future. 

## 4. Conclusions

In conclusion, considering that the worldwide laws and regulations that restrict the content of Pb in functional materials are about to be implemented, and that the currently reported Pb-free piezoelectric materials are far from commercial fabrication, we propose to design low-Pb high-piezoelectric material as an effective alternative. Based on the MPB mechanism, a new MPB-involved system, BZ_0.18_T-xBP_0.22_T, was fabricated. The optimum composition (also the MPB composition), x = 0.5, exhibited a high piezoelectricity of 240 pC/N and a high Tm of 121.1°C. Our result verifies the validity of obtaining good piezoelectricity via fabricating a low-Pb MPB system with one end member of a Pb-free compound and the other end member of a Pb-based compound. We believe that low-Pb piezoelectric materials that can completely replace Pb-rich materials are coming soon. 

## Figures and Tables

**Figure 1 materials-15-04760-f001:**
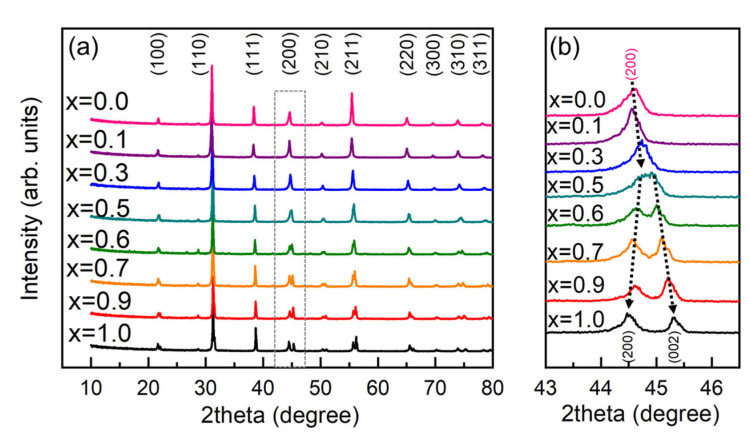
XRD profiles (**a**) and the (200) reflections (**b**) of the BZ_0.18_T–xBP_0.22_T ceramic powder samples at room temperature (22°C).

**Figure 2 materials-15-04760-f002:**
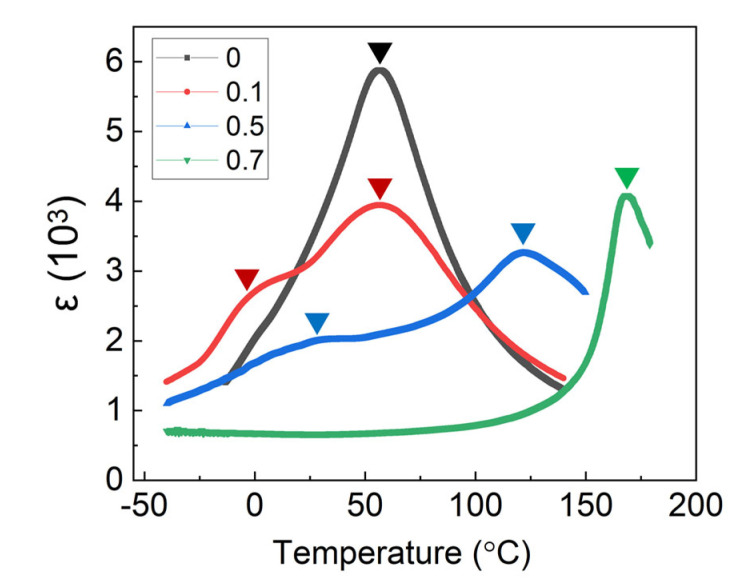
Dielectric permittivity versus temperature curves (10 kHz) for the selected compositions x = 0, 0.1, 0.5, and 0.7 of BZ_0.18_T–xBP_0.22_T.

**Figure 3 materials-15-04760-f003:**
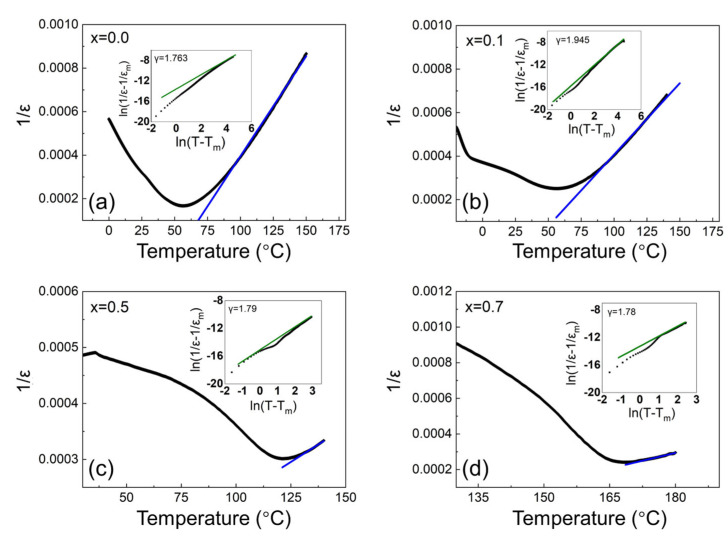
The inverse dielectric permittivity (10 kHz) versus temperature curves and corresponding fitted ln($\frac{1}{\varepsilon }-\frac{1}{\varepsilon_{m}}$) versus ln($T-T_{m}$) curves (inset) for the selected compositions x = 0 (**a**), 0.1 (**b**), 0.5 (**c**), and 0.7 (**d**) of BZ_0.18_T-xBP_0.22_T. The blue lines denote the linear fitting of 1/ε~T above T_C_, and the green lines denote the linear fitting of ln($\frac{1}{\varepsilon }-\frac{1}{\varepsilon_{m}}$) ~ ln($T-T_{m}$) at high temperature regime.

**Figure 4 materials-15-04760-f004:**
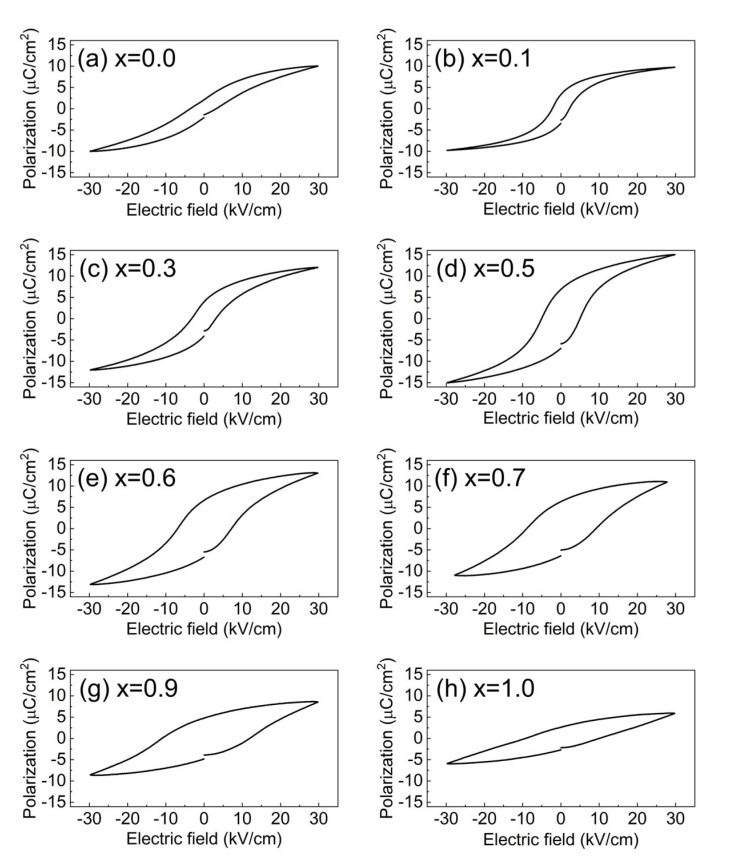
Polarization (P)—electric field (E) hysteresis loops for BZ_0.18_T–xBP_0.22_T: (**a**) 0, (**b**) 0.1, (**c**) 0.3, (**d**) 0.5, (**e**) 0.6, (**f**) 0.7, (**g**) 0.9, and (**h**) 1.0.

**Figure 5 materials-15-04760-f005:**
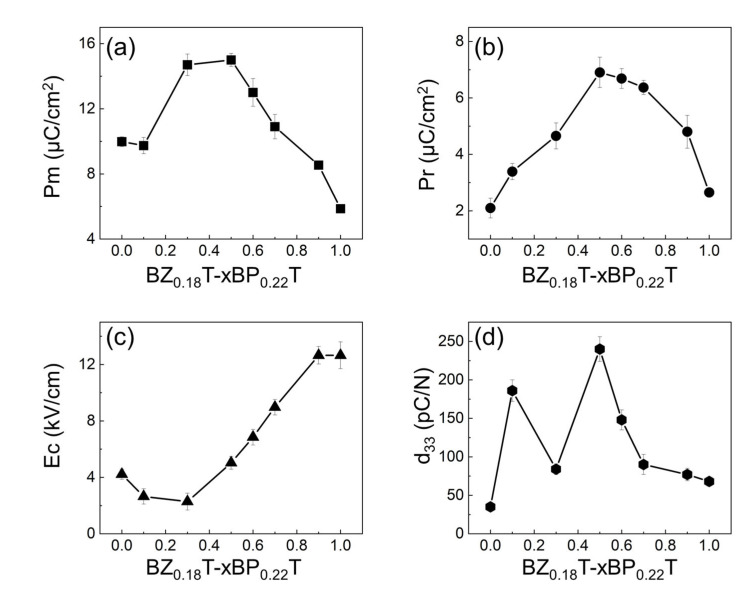
Composition dependence of the maximum polarization Pm (**a**), remnant polarization Pr (**b**), coercive field Ec (**c**), and the piezoelectric coefficient d_33_ (**d**) for BZ_0.18_T–xBP_0.22_T. The lines in the figure are guide for the eye.

**Table 1 materials-15-04760-t001:** The dielectric parameters of BZ_0.18_T–xBP_0.22_T (x = 0.0, 0.1, 0.5, 0.7) at 10 kHz.

Composition	T_C_ (°C)	T_m_ (°C)	C (°C)	γ
0.0	48.8	56.4	3.63 × 10^6^	1.763
0.1	54.6	56.0	1.14 × 10^7^	1.945
0.5	85.0	121.1	6.10 × 10^6^	1.790
0.7	130.0	168.6	1.21 × 10^6^	1.780

## Data Availability

Not applicable.

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
