# Peer review of "Low-Pb High-Piezoelectric Ceramic System (1−x)Ba(Zr0.18Ti0.82)O3–x(Ba0.78Pb0.22)TiO3"

_materials, 2022, doi:10.3390/ma15144760_

Round 1

Reviewer 1 Report

The manuscript entitled "Low-Pb high-piezoelectric ceramic system (1-x)Ba(Zr0.18Ti0.82)O3- 2

x(Ba0.78Pb0.22)TiO3" written by Zhou C. et al. is a very interesting article about the use of low Pb content in piezoelectric materials. However, in order to accept this work, there are some aspects that authors must achieve:

·         Could authors explain why 0.22 Pb content was selected?

·         It is known that piezoelectrical properties are influenced by the MPB and grain size. However, authors don’t show the influence of grain size toward the increase of Pb content. It should be interesting if authors could abord not only the crystalline aspects but also microstructural and morphological characteristics of the ceramics.

·         Using the Curie Weiss Law authors calculated the degree of diffusion constant (γ). It seems that an optimum relaxor behavior is achieved for 0.1 ceramics (Figure 3). How is this associated to crystalline characteristics?

·         Figure 2 shows the dielectric permittivity of x=0,0.1, 0.5 and 0.7 ceramics. What happened to dielectric losses?

·         Authors show a composition dependence between different ferroelectric characteristics (Pr, Pm, Ec) and the piezoelectric coefficient d33 (Figure 5). However , for any device design, the electromechanical coupling factor (kp) as well as dielectric looses (tan d) are important properties to be considered. If possible, could authors complete with some other piezoelectric coefficients?

Reviewer 2 Report

The article is devoted to the actual problem of developing new low-Pb high-performance piezoelectric and ferroelectric materials. The article is well prepared and I believe that it can be published after minor revisions.

1. Unfortunately, the authors limited themselves to only a frequency of 10 kHz in the study of the properties of materials. Since piezo and ferroelectric materials have potential applications at higher frequencies, it would be useful and interesting to show their characteristics in a wider frequency range.

2. Readers would benefit from more physical interpretation of the results obtained. Now they are explained rather sparingly. Perhaps it would be useful to add a Discussions section.

3. It has the meaning to note that lead has an increased migration ability during the formation of ferroelectric materials with their content, which can affect the properties and microstructure (See for example https://doi.org/10.1134/S108765961402014X; https://doi.org/10.1002/sia.6255 and https://doi.org/10.3390/ma12182926). What is the role of the Pb induced non-stoichiometric and imperfection factors in your materials?

Round 2

Reviewer 1 Report

All comments were addressed correctly.